# Development and Application of a Physiologically Based Pharmacokinetic Model for Diclazuril in Broiler Chickens

**DOI:** 10.3390/ani13091512

**Published:** 2023-04-29

**Authors:** Fang Yang, Mei Zhang, Yang-Guang Jin, Jun-Cheng Chen, Ming-Hui Duan, Yue Liu, Ze-En Li, Xing-Ping Li, Fan Yang

**Affiliations:** College of Animal Science and Technology, Henan University of Science and Technology, Luoyang 471023, Chinajinyangguang1999@126.com (Y.-G.J.);

**Keywords:** physiologically based pharmacokinetic (PBPK) model, withdrawal interval, diclazuril, broiler chickens, residue prediction

## Abstract

**Simple Summary:**

Coccidiosis is a devastating poultry disease with significant economic implications. The prevention and control of coccidiosis currently rely on the administration of anticoccidial drugs through feed or water. One of the commonly used drugs is diclazuril, which was approved in broilers for their whole life. Additionally, its prolonged use can lead to residue accumulation in edible tissues. As a result, we developed and validated a physiologically based pharmacokinetic (PBPK) model for diclazuril to predict its residues in broilers. We identified and analyzed the key parameters associated with diclazuril concentrations in the muscle through Monte Carlo analysis. Our findings suggested that a withdrawal period of 0 days was suitable for both recommended dosing regimens. This model can be expanded to other coccidiostats and poultry species as a critical resource.

**Abstract:**

Withdrawal periods for diclazuril in broilers have traditionally been determined through regression analysis. However, over the last two decades, the physiologically based pharmacokinetic (PBPK) model has gained prominence as a predictive tool for veterinary drug residues, which offers an alternative method for establishing appropriate withdrawal periods for veterinary drugs. In this current study, a flow-limited PBPK model was developed to predict diclazuril concentrations in broilers following long-duration administration via medicated feed and water. This model consists of nine compartments, including arterial and venous plasma, lung, muscle, skin + fat, kidney, liver, intestine contents, and the rest of the body compartment. Physiological parameters such as tissue weights (V_cxx_) and blood flow (Q_cxx_) were gathered from published studies, and tissue/plasma partition coefficients (P_xx_) were calculated through the area method or parameter optimization. Published diclazuril concentrations were compared to the predicted values, indicating the accuracy and validity of the model. The sensitivity analysis showed that parameters associated with cardiac output, drug absorption, and elimination significantly affected diclazuril concentrations in the muscle. Finally, a Monte Carlo analysis, consisting of 1000 iterations, was conducted to calculate the withdrawal period. Based on the Chinese MRL values, we calculated a withdrawal period of 0 days for both recommended dosing regimens (through mediated water and feed at concentrations of 0.5–1 mg/L and 1 mg/kg, respectively). However, based on the European MRLs, longer periods were determined for the mediated feed dosing route. Our model provides a foundation for scaling other coccidiostats and poultry species.

## 1. Introduction

Coccidiosis is a common and serious poultry disease caused by the genus *Eimeria* which affects the intestinal tract of birds [1]. Chickens affected by this disease typically exhibit symptoms such as diarrhea and listlessness. These symptoms are further compounded by malabsorption, which can result in reduced feed conversion efficiency and significant economic losses for the poultry industry worldwide [2]. Currently, the prevention and control of coccidiosis mainly depend on the addition of anticoccidial drugs in feed and water [1].

Diclazuril is an anticoccidial drug with high effectiveness used in major poultry species such as broilers, pullets, and turkeys, as well as in minor edible bird species [3]. Its action mechanism remains unclear, but diclazuril effectively hinders the asexual and sexual growth of coccidia by impeding the excretion of oocysts, which ultimately halts the lifecycle of these parasites. In China, the available preparations of diclazuril are solution and premix, which are recommended for use in chickens through drinking water (0.5–1 mg/L) and mixed into feed (1 mg/kg), respectively [4]. This drug is approved for use in broilers throughout the growing cycle. However, it is not intended for use in laying hens.

The application of diclazuril throughout the whole growing cycle may cause potentially harmful drug residues in chicken. In rodents, the toxicological effects of diclazuril were mainly observed in the central nervous system [5]. Vomiting and defecation were also observed in dogs [5]. Although no similar toxicological reports exist for humans, the potential risks of residue ingestion make it imperative to closely monitor diclazuril levels in broilers. In China, the maximum residue limits (MRLs) of diclazuril in broiler products differ across various tissue types, which are 500, 1000, 2000, and 3000 μg/kg in muscle, skin + fat, kidney, and liver [6]. However, lower MRL values were labeled in Europe with the corresponding values at 500, 500, 1000, and 1500 μg/kg, respectively [7]. To ensure food safety, the Chinese broiler industry has implemented a 5-day withdrawal period for both diclazuril premix (1 mg/kg) and solution (0.5–1 mg/L) that are mixed with the feed and dissolved in water, respectively.

In recent years, physiologically based pharmacokinetic (PBPK) models have emerged as a reliable tool for forecasting veterinary drug residues in animal-derived foods such as meat [8,9,10], milk [11,12,13], and eggs [14,15,16]. Compared to traditional post-slaughter monitoring methods, PBPK models are predictive and work based on mass-balance equations that are defined by physiological mechanisms [8]. Previous PBPK models in chickens were mostly developed for antibacterial drugs [10,17], with only one model available for the more widely used coccidiostats (monensin) [18]. Unfortunately, there is no available model for diclazuril, making it imperative to establish a PBPK model to forecast diclazuril residues in broilers. Given that the disposition characteristics of various compounds differ significantly, this study seeks to develop a PBPK model to aid in predicting diclazuril levels in broilers and to further calculate the withdrawal periods of diclazuril.

## 2. Materials and Methods

### 2.1. Concentrations versus Time Data

We obtained diclazuril concentrations in plasma and tissue from published reports [1,19,20,21,22], using tables directly or figures extracted with GetData Graph Digitizer (version 2.26; http://getdata-graph-digitizer.com; accessed on 10 March 2023). See Table 1 for key details on selected studies.

### 2.2. Model Structure

The current PBPK model (Figure 1), which consists of nine compartments, including arterial plasma, venous plasma, lung, muscle, skin + fat, kidney, liver, intestine contents, and the rest of the body compartment, was adapted from a previous broiler model [10]. Four of the compartments serve as target tissues (muscle, skin + fat, kidney, and liver), three are the connective ones (blood and lung), one is the site of diclazuril exposure (intestinal contents), and one is a virtual mixing compartment. Consistent with the previous broiler model, diclazuril was assumed to be flow-limited.

Diclazuril is commonly administered to chickens through medicated water and feed. To model how the drug is absorbed, we developed an intestine compartment (Figure 1). To simplify this current model, neither crop nor the absorption time lag was included in it. Additionally, we assumed that diclazuril entered the gut directly and the intestine tissue lacked blood flow, which acted solely as the site of drug exposure and absorption. In previous studies [1,19,20,22], chickens were provided with medicated feed or water ad libitum. As chickens commonly cease consuming food and water in the dark, the previously reported light regimes were utilized to simulate drug exposure. To further simplify this model, we assumed that chickens consumed 1.1 kg of feed and 0.5 L of water per day, evenly distributed during light hours. Additionally, after dosing, diclazuril was assumed to be immediately available in the intestinal tract for absorption at a rate constant (K_a_; h^−1^). The bioavailability of diclazuril is not known, although it has been noted that absorption is limited [23]. Therefore, we utilized this current model to accurately calculate its bioavailability as the ratio of total absorption to total dosage. More details can be found in the model code provided in the Appendix A. Any unabsorbed diclazuril was eliminated through feces at a rate of K_gut_ (h^−1^). Furthermore, hepatic elimination was simulated using the parameter of Cl_he_ (L/h/kg).

Diclazuril, upon absorption, was transported through the bloodstream to various tissues. Mass-balance differential equations were employed to describe the diclazuril change rate in each compartment (Table 2), as described previously [10,17]. Studies have shown that in different species—including chickens—diclazuril is primarily eliminated through fecal excretion, with little excretion in urine [23]. Furthermore, metabolites accounted for less than 6% of the total dose in various species [23]. Therefore, it can be inferred that diclazuril is primarily eliminated with feces, followed by hepatic metabolism. Additionally, the parameters of K_gut_ and Cl_he_ were used to simulate the elimination of diclazuril (Figure 1). More details about the differential equations describing diclazuril absorption, distribution, and elimination can be found in Table 2.

It was assumed that all diclazuril was immediately available in the intestinal tract after dosing, and from there, diclazuril was absorbed with the rate constant of K_a_ (h^−1^). The unabsorbed diclazuril was eliminated with feces at the rate of K_gut_ (h^−1^). The t is for time (h). Cl_he_ is the hepatic clearance (L/h/kg) of diclazuril. Q_tot_ is cardiac output (L/h), which is equal to the plasma flow through the lung.

### 2.3. Model Parameterization

The physiological and anatomical parameters in broilers were derived from a literature review [24]. Tissue weight and blood flow (Table 3) were denoted as V_cxx_ and Q_cxx_, respectively, and were expressed as a percentage of total body weight (BW) and cardiac output (CO). Due to the absence of growth data on various chicken breeds, we assumed that the body weight of broilers remained consistent throughout the treatment and simulation period. Partition coefficients (P_xx_s; Table 3) for diclazuril in non-eliminating tissues were calculated using the area method [25] based on previously reported tissue and plasma concentrations [20]. Hematocrit, abbreviated as pcv, was utilized to correlate blood and plasma volume, which was assumed to be 32 ± 2.76% [24] (more details can be found in the model code provided in the Appendix A).

Diclazuril was deemed not to enter blood cells to simplify the present simulation, and the amount of diclazuril was assumed to be equal in both blood and plasma. Because the average BWs of chicken varied among previously published studies (Table 1), the present model was adjusted according to those specific BWs. According to the previous study [24], the value of CO was 9.88 L/h/kg BW and was converted to L/h via the following equation: Q_tot_ = CO × BW. Due to the lack of concentration-time data for diclazuril in the lung and rest compartment, the area method could not be used for calculating the values of P_lu_ and P_re_. Hence, optimization using the plasma and tissue concentration data [1,19,21] helped determine the P_xx_ values in the lung, liver (eliminating tissue), and the rest compartment.

Apart from the optimized P_xx_s mentioned above, we had difficulty obtaining values for the other three parameters, including K_a_, K_gut_, and Cl_he_. Thus, we used the published diclazuril concentrations (Table 1) to optimize them. We performed all parameter optimizations in acslxtreme software, using the Nelder–Mead maximum likelihood algorithm [26].

The current model was developed using acslxtreme software (version 3.0.2.1; Aegis Technologies Group, Inc., Huntsville, Alaska). All simulations, optimizations, validations, sensitivity analyses, and Monte Carlo analyses were also performed using this software. Further details can be found below.

### 2.4. Model Validation

To validate the model, we visually compared the predicted concentrations in plasma and tissues with previously published observations without using any actual observations during the parameter optimization process. We also performed linear regression analysis and calculated the correlation coefficient. Additionally, we calculated the mean absolute percentage error (MAPE) using the following equation to evaluate the validity of the current PBPK model: $MAPE\left(\%\right)=\frac{1}{N}\sum_{i=1}^{N}\frac{\left|C_{oi}-C_{pi}\right|}{C_{oi}}\times 100\%$, where N stands for the number of observations, and C_oi_ and C_pi_ represent the observed and predicted diclazuril concentrations, respectively [27]. The evaluation criteria for MAPE are as follows: acceptable prediction (MAPE < 50%), good prediction (10% < MAPE < 20%), and excellent prediction (MAPE < 10%).

### 2.5. Sensitivity Analysis

According to our previous reports [8,27], a local sensitivity analysis was performed to determine which parameters had the most influence on the diclazuril concentrations in muscle, and the normalized sensitivity coefficient (NSC) values were calculated for each parameter [27]. A parameter was considered significant during postexposure if its NSC exceeded an absolute value of 0.25 [28]. To simultaneously perform the sensitivity analysis for all parameters used in both routes, single exposure through mediated feed (0.5 mg/kg) and water (0.5 mg/L) was simulated simultaneously, and both exposures lasted 12 h.

### 2.6. Monte Carlo analysis and Withdrawal Interval Estimation

Following the sensitivity analysis, influential parameters were subjected to Monte Carlo analysis to generate a virtual population of broilers (n = 1000) [26,29]. Central tendency and spread were determined using a previous report [24] or parameter optimization, as detailed in the above model parameterization. As suggested by previous research [10], normal distribution was assumed for these parameters, and distribution information is shown in Table 4.

Monte Carlo analysis was performed using acslxtreme software to generate one thousand iterations (n = 1000). Each simulation randomly generated parameter values with mean, standard deviation (SD), lower bound (mean − SD), and upper bound (mean + SD), which were integrated into our model to predict diclazuril concentrations. We used 2 recommended levels of diclazuril addition: medicated feed containing 1 mg/kg diclazuril and water containing 1 mg/L diclazuril, for 4 different dosing durations (5, 10, 15, and 20 consecutive days) for each route of administration. Additionally, two specific dosing regimens were simulated for validation studies [20,22]. Additionally, the published light regimes (Table 1) were used for drug exposure in each simulation.

Predicted diclazuril concentrations were generated after each run, creating 1000 virtual individuals. Withdrawal intervals were calculated to ensure that the predicted concentrations remained below both Chinse and European MRL values. Additionally, both recommended methods [26,30] were used to calculate withdrawal periods so that diclazuril concentrations in the 95th percentile of the population (n = 1000) were below MRL values.

## 3. Results

### 3.1. Model Parameters

In Table 3, we present the physiological and anatomical parameters used in our model, which were derived from a prior review [24]. To obtain the partition coefficient for diclazuril, we utilized the area method or parameter optimization. The final values of partition coefficients were as follows: 0.0955 for skin + fat, 0.1299 for muscle, 0.5603 for lung, 0.6813 for kidney, 0.9613 for liver, and 1.2965 for the rest compartment (Table 3). In addition, we determined the values for parameters related to absorption and elimination through optimization, and they are listed below: K_a_ = 0.1234 h^−1^; K_gut_ = 0.3838 h^−1^; Cl_he_ = 0.00344 L/h/kg.

The absolute bioavailability of diclazuril is unknown. To calculate it, we determined the ratio of the absorbed amount to the total administered dose. As the models with both routes had the same K_a_ and K_gut_, the absolute bioavailability after dosing through both feed and water was calculated as 24.32%.

### 3.2. Model Validation

To validate the model, predicted and previously published diclazuril concentrations were visually compared (Figure 2 and Figure 3). The figures demonstrate accurate predictions, with most of the predicted concentrations closely matching reported values at the corresponding time points. Linear regression analysis was also performed (Table 5). It was shown that the present model was generally acceptable because most of the determination coefficients were higher than 0.75 (Table 5), which indicated generally acceptable results with determination coefficients higher than 0.75—a criterion previously established [13]. To validate the model, MAPEs were also calculated, and values ranged from 2.94% to 16.97%, confirming excellent and good predictions for all simulations [27].

### 3.3. Sensitivity Analysis

Figure 4 revealed the most significant parameters affecting diclazuril concentrations in muscle as determined via local sensitivity analysis. The parameter Q_clu_ had the largest impact, aligning with the flow-limited model assumption. Additionally, absorption and elimination parameters were found to be influential, with Cl_he_ being the most impactful, followed by K_gut_ and K_a_. Concerning P_xx_s parameters, only P_mu_ and P_re_ had any noteworthy effect on muscle concentrations. Like Q_cxx_s (Q_cmu_) and V_cxx_s (V_cmu_, V_csk_, and V_cbl_), these parameters remained influential throughout the simulation with |NSC| > 0.25. However, certain parameters like BW, Q_cmu_, K_a_, and K_gut_ had constant NCS values for the whole simulation period or some specific periods, indicating consistent influence. These findings suggest that certain parameters have inconsistent impacts, while others have consistent impacts, though all of them are considered sensitive. This study found that only Cl_he_ had a significant impact on diclazuril concentrations in muscle, while other liver parameters did not. The results for non-sensitive parameters are shown in Appendix A.

### 3.4. Monte Carlo Analysis and Withdrawal Interval Estimation

We used Monte Carlo analysis to predict diclazuril concentrations after two recommended treatment routes at four durations (Appendix A). Our results showed steady-state drug concentration in all tissues after 15 days of continuous feeding or drinking (Appendix A). All predictions in 1000 virtual individuals were below the corresponding Chines MRL values (Appendix A). Consequently, based on the Chinese MRLs, we recommend a withdrawal period of 0 days for both dosing regimens. For the European MRL values, we arrived at the same recommendation of a 0-day withdrawal period following four different dosing durations through mediated water. However, a longer withdrawal period was calculated for the mediated feed administration route. Specifically, after 5, 10, 15, and 20 days of continuous feeding, the withdrawal periods were calculated as 0.63, 1.41, 1.63, and 1.65 days, respectively.

We also simulated two dose regimens from previous studies [20,22] (see Appendix A). Additionally, the results indicated that a mediated concentration of 3 mg/L in water resulted in higher residues in the kidney and liver that exceeded both MRL values (see Appendix A). Accordingly, we calculated withdrawal periods for this regimen [20]. The current results, based on both Chinese and European MRL values, differ from the previous linear regression result (3 days). The current withdrawal periods are shorter (1 day) and longer (4 days).

## 4. Discussion

This study introduces the first PBPK model for diclazuril in broilers. While another team of researchers previously developed a similar model for the anticoccidial drug monensin [18], our model includes compartments for the lung, kidney, and skin (expressed as skin + fat). Furthermore, we opted for a single virtual rest compartment rather than the two richly and poorly perfused compartments used in the earlier model. In contrast to the previous model, which relied exclusively on liver pathways (metabolism and biliary excretion), our updated model includes hepatic metabolism and intestinal excretion due to the diverse dispositions of the compounds studied. Despite their differences, both models adhered to the blood-flow limited assumption and were created using acslxtreme software. They were validated through external data sets and underwent sensitivity analysis.

The P_XX_ values of diclazuril were determined through the area method [25] or parameter optimization in various tissues. Results in skin + fat, muscle, lung, kidney, liver, and other compartments were 0.0955, 0.1299, 0.5603, 0.6813, 0.9613, and 1.2965, respectively (refer to Table 3). Conversely, the previous PBPK model showed that monensin had P_XX_ values of 0.51, 0.83, and 3.39 in muscle, liver, and fat, respectively [18]. These findings shed light on the selective tissue distribution of both anticoccidial drug drugs, emphasizing that diclazuril exhibits lower fat affinity compared to monensin.

We obtained the values of absorption and elimination parameters through optimization, which yielded final values of 0.1234 h^−1^, 0.3838 h^−1^, and 0.00344 L/h/kg for K_a_, K_gut_, and Cl_he_. These results indicate that diclazuril has a slow absorption rate but a fast excretion rate, mostly through feces. This finding is consistent with previous research [23]. Based on the current model, it appears that there may be limited absorption for diclazuril when dosing through both mediated feed and water, with a calculated absolute bioavailability of 24.32%. However, when dosed through feed, the bioavailability of monensin appears to be much lower (3.9%) [18] than diclazuril. It is recommended to conduct further in vivo experiments to verify the absolute bioavailability of diclazuril and monensin.

In the previous model for monensin [18], the crop compartment was incorporated, and a lag time was simulated. Initially, we tried to do the same in the current model, but we discovered that it had poor predictive ability. Moreover, the lag time varied considerably among individuals. Therefore, we excluded the crop compartment and lag time in the final model, which might have also contributed to an artifact of slow absorption for diclazuril. However, in a previous study [1], gavage administration of a single dose to broilers resulted in peak concentrations at 29.1 to 34.3 h, which also demonstrated slow absorption of diclazuril.

The validation results showed that our model could accurately predict diclazuril concentrations in various tissues after administration through feed or water. As shown in Figure 2 and Figure 3, excellent coverage was achieved for all tissues and plasma. The linear regression analysis and MAPE results have also proved good or excellent predictions in all samples.

In this study, a local sensitivity analysis was conducted to identify the influential parameters affecting diclazuril concentrations in muscles. The results indicated that the Q_clu_ parameter was the most influential, in line with the flow-limited model assumption. Other parameters related to absorption and elimination were also found to be significant. Notably, K_a_ and K_gut_ only impacted the exposure phase (<24 h), while Cl_he_ was influential during the entire simulation period. Interestingly, the muscle-related parameters (V_cmu_, P_mu_, and Q_cmu_) exhibited varying degrees of influence; V_cmu_ had the most impact, followed by P_mu_ and Q_cmu_. After approximately 110 h, V_cmu_ had a negative effect, but the underlying cause is unknown. Therefore, an in vivo pharmacokinetic study is recommended to explore potential explanations. Throughout the simulation, the BW parameter remained fixed, which aligns with the fixed body weight assumption.

The fixed body weight was one imperfection of the present model. Fortunately, the ages of broilers used in the model were all close to the market age [24], and their body weights were deemed relatively consistent. To simulate lifetime exposure, longer exposure durations were employed. Nonetheless, diclazuril reached a steady state after 15 days of continuous feeding or drinking (Appendix A), suggesting exposure of up to 20 days would suffice. Future studies could develop a lifetime PBPK model [31] for diclazuril in young chicks while incorporating variable body weight.

We utilized a Monte Carlo analysis to predict diclazuril concentrations in a large population (n = 1000) and account for interindividual variability. These predictions were then compared to both Chinese and European MRL values in different tissues. For both recommended dosing regimens, the predictions in all 1000 virtual individuals were lower compared with the Chinese MRL values, including the liver (a tissue with commonly high residues). As a result, the withdrawal period for diclazuril should be 0 days based on the Chinese MRL values. However, based on the European MRL values, longer periods were determined for the mediated feed dosing route. Based on the Monte Carlo analysis, a higher concentration of diclazuril in water (3 mg/L) [20] was also simulated. The results demonstrated that a withdrawal period of 1 day and 4 days was sufficient to ensure diclazuril concentrations in the 95th percentile remained below the Chinese and European MRL values, respectively. All these results suggested that the CCVP’s guidance value (5 days) [4] has minimal risk.

## 5. Conclusions

Our research successfully developed and validated a PBPK model for diclazuril residues in broilers. The model accurately predicted diclazuril concentrations after continuous administration through medicated feed and water for different durations. We determined the influential parameters on muscle concentrations and conducted further Monte Carlo analysis. As a result, based on the Chinese MRL values, we calculated a withdrawal period of 0 days for both recommended dosing regimens. However, based on the European MRLs, longer periods were determined for the mediated feed dosing route. This model can serve as a foundation for scaling other coccidiostats and poultry species. The current results proved that this PBPK model can serve as an alternative to determine the withdrawal period for diclazuril in broilers. However, further studies on pharmacokinetics and depletion in younger chicks are essential to validate and apply this model.

## Figures and Tables

**Figure 1 animals-13-01512-f001:**
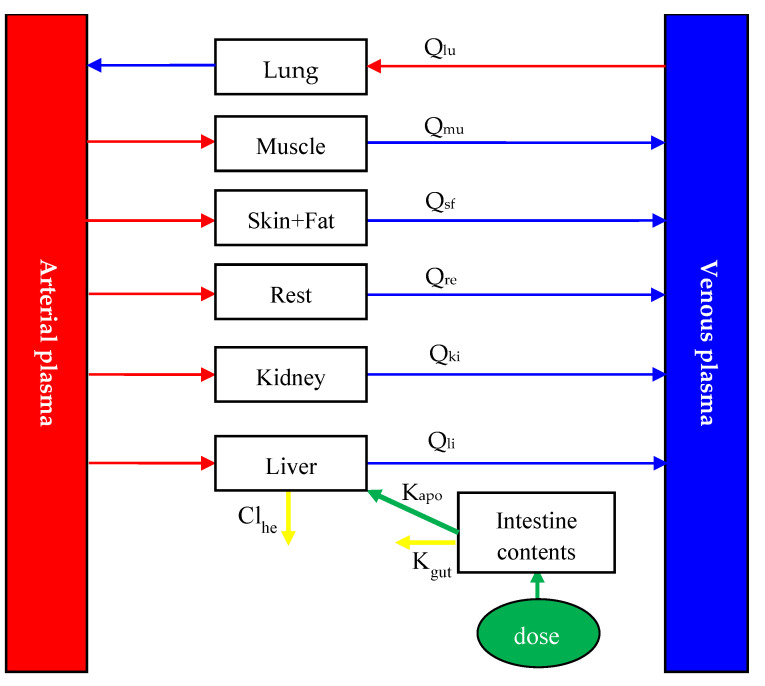
Schematic diagram of the physiologically based pharmacokinetic model for diclazuril in broiler chickens after oral administration. Q_xx_ (L/h) is plasma flow through some tissue. Subscript xx is the name of tissue, and lu, mu, sf, li, ki, and re were abbreviations for lung, muscle, skin + fat, liver, kidney, and the rest of the body compartment, respectively. Based on previous studies [1,19,20,22], diclazuril was orally given to chickens through medicated feed or water. It was assumed that all diclazuril was immediately available in the intestinal tract after dosing, and from there, diclazuril was absorbed with the rate constant of K_a_ (h^−1^). The unabsorbed diclazuril was eliminated with feces at the rate of K_gut_ (h^−1^). In addition to intestinal elimination, the parameter of Cl_he_ (L/h/kg) was used to simulate the hepatic elimination of diclazuril.

**Figure 2 animals-13-01512-f002:**
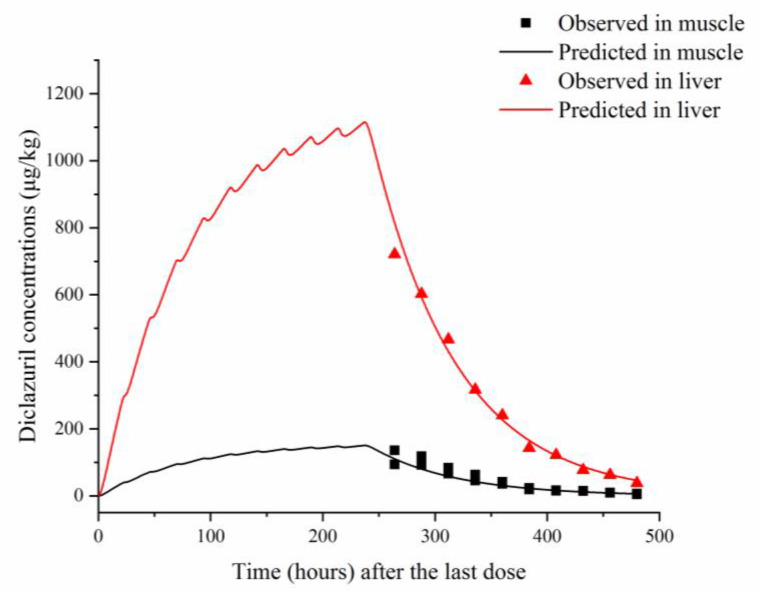
Comparisons between predicted (curves) and published (points; [22]) diclazuril concentrations (μg/kg) in tissues after 10 consecutive days of administering medicated feed containing 730 μg/kg of diclazuril.

**Figure 3 animals-13-01512-f003:**
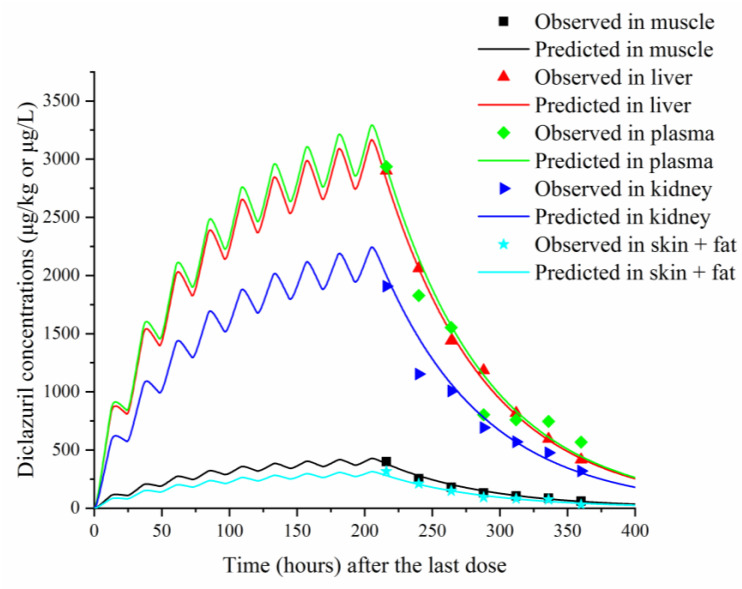
Comparisons between predicted (curves) and published (points; [22]) diclazuril concentrations (μg/kg or μg/L) in tissues and plasma after 9 consecutive days of administering medicated water containing 3 mg/kg of diclazuril.

**Figure 4 animals-13-01512-f004:**
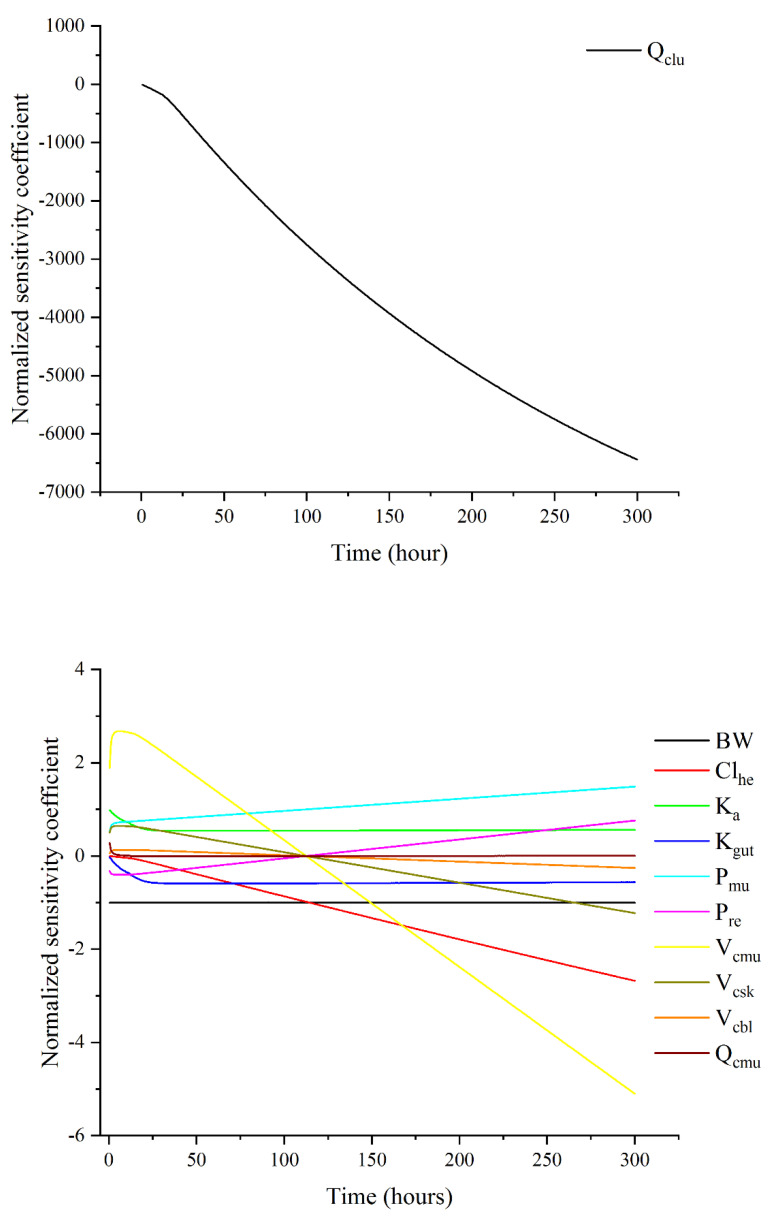
Sensitivity analysis results for those influential parameters on diclazuril concentrations in muscle.

**Table 1 animals-13-01512-t001:** Pharmacokinetic studies used in model optimization and validation.

Ref. #	Purpose ^1^	Routes and Dose	Light Regime (Light/Dark) ^2^	BW (kg)	Age (Days)	Breeds	Matrix ^3^
[1,19] ^4^	Optimization	Single oral dose at 1 mg/kg BW by gavage ^5^	12/12	1.52	50	Lingnan Yellow Chicken	PL
		Ate medicated feed at 1 mg/kg for 7 consecutive days					MU, SF, LI, KI
[20]	Validation	Drunk medicated water at 3 mg/L for 9 consecutive days	12/12	1.34	30	Lingnan Yellow Chicken	PL, SF, MU, LI, KI
[22]	Validation	Ate medicated feed at 730 μg/kg for 10 consecutive days	21/3	1.5	21	Ross 308	MU ^6^, LI
[21]	Optimization	Single oral dose at 80 μg/kg BW by gavage	18/6	1.23	15	Ross 308	PL

^1^ This current model utilized five sets of concentration-time data. Three sets were used for parameter optimizations, while two were used for validation purposes. ^2^ Light regime was related to the duration of daily exposure in multiple-dose treatments because chickens stop drinking or eating in the dark. ^3^ The abbreviations for matrix: PL: plasma; MU: muscle; LI: liver; KI: kidney;. ^4^ In these studies, both enantiomeric forms (R- and S-) of diclazuril were quantified. We use their sum for parameter optimization. ^5^ The current PBPK model was developed by CSL encoding and implemented using acslX software (version 3.0.2.1). Additional information on the code can be found in the Appendix A. To simulate multiple sustained exposures to diclazuril, we utilized the PULSE function. However, this model could also reflect a single gavage by modifying the model with a last exposure day (Dstop) of 1 day and exposure time (tlen) of 0.001 h. Therefore, the diclazuril concentrations in plasma after one single dose by gavage were also used to optimize the model parameters. ^6^ These data sets in muscle (MU) included the breast and thigh.

**Table 2 animals-13-01512-t002:** Differential equations describing the change rate of diclazuril amount (μg) or concentrations (μg/kg or μg/L) in each compartment.

Compartment	Differential Equation
Intestinal contents	$\frac{dA_{ic}}{dt}=\frac{dose}{tlen}-K_{a}\times A_{ic}-K_{gut}\times A_{ic}$
Muscle	${V_{mu}\times \frac{dC_{mu}}{dt}=Q}_{mu}\times \left(C_{ap}-\frac{C_{mu}}{P_{mu}}\right)$
Skin + fat	$V_{sf}\times {\frac{dC_{sf}}{dt}=Q}_{sf}\times \left(C_{ap}-\frac{C_{sf}}{P_{sf}}\right)$
Kidney	$V_{ki}\times {\frac{dC_{ki}}{dt}=Q}_{ki}\times \left(C_{ap}-\frac{C_{ki}}{P_{ki}}\right)$
Liver	$V_{li}\times \frac{dC_{li}}{dt}=K_{a}\times A_{ic}+Q_{li}\times \left(C_{ap}-\frac{C_{li}}{P_{li}}\right)-{Cl}_{he}\times \frac{C_{li}}{P_{li}}$
Lung	$V_{lu}\times {\frac{dC_{lu}}{dt}=Q}_{tot}\times \left(C_{vp}-\frac{C_{lu}}{P_{lu}}\right)$
Rest	${V_{re}\times \frac{dC_{re}}{dt}=Q}_{re}\times \left(C_{ap}-\frac{C_{re}}{P_{re}}\right)$
Arterial plasma	$V_{ap}\times {\frac{dC_{ap}}{dt}=Q}_{tot}\times \left(\frac{C_{lu}}{P_{lu}}-C_{ap}\right)$
Venous plasma	$V_{vp}\times {\frac{dC_{vp}}{dt}=Q}_{re}\times \frac{C_{re}}{P_{re}}+Q_{li}\times \frac{C_{li}}{P_{li}}+Q_{ki}\times \frac{C_{ki}}{P_{ki}}+Q_{sf}\times \frac{C_{sf}}{P_{sf}}\;\;\;\;\;\;\;+Q_{mu}\times \frac{C_{mu}}{P_{mu}}-Q_{tot}\times C_{vp}$

A_xx_ and C_xx_ are the amount (μg) and concentration (μg/kg or μg/L) of diclazuril in each compartment, respectively. V_xx_ and Q_xx_ are the volume (L) and plasma flow (L/h) through a tissue, respectively, whereas P_xx_ is the partition coefficient (unitless) for diclazuril in a tissue. Subscript xx is the name of a compartment, and ic, mu, sf, ki, li, lu, ap, vp, and re are abbreviations for intestinal contents, muscle, skin + fat, kidney, liver, lung, arterial plasma, venous plasma, and the rest of the body compartment, respectively. In this present model, diclazuril was given to chickens through medicated feed or water. The parameter of dose represents the dose of diclazuril (μg), which was calculated by multiplying the daily feed or water intake by the diclazuril concentration in the feed or water. Additionally, each chicken had free access to medicated feed and water. It is well known that chickens stop drinking and eating in the dark. Therefore, the parameter of tlen was used to simulate the diclazuril exposure time (hour) per day.

**Table 3 animals-13-01512-t003:** Parameters of tissue weights, plasma flows, and tissue/plasma partition coefficients used in the current physiologically based pharmacokinetic model.

Compartment	Tissue Weight ^1^ (V_cxx_, Fraction of BW)	Blood Flow ^2^ (Q_cxx_, Fraction of Cardiac Output)	Partition Coefficient for Diclazuril ^3^ (P_xx_)
Muscle	0.5712	0.0764	0.1299
Skin + fat	0.2678	0.2505	0.0955
Kidney	0.0064	0.2012	0.6813
Liver	0.0214	0.2526 ^4^	0.9613 ^5^
Lung	0.0071	1 ^6^	0.5603 ^5^
Arterial plasma	0.0322	NA	NA
Venous plasma	0.0161	NA	NA
Rest	0.0778 ^7^	0.2193 ^8^	1.2965 ^5^

^1^ The current PBPK model and all simulations were adjusted based on the chicken’s body weights (BWs). The weights of different tissues were expressed as the fractions of BW. Due to the absence of growth data on various chicken breeds, we assumed that the body weight of broilers remained consistent throughout the treatment and simulation period. ^2^ The cardiac output (CO) was derived from a review [24], whose final value was 9.88 L/h/kg BW. Additionally, all plasma flows through different tissues were expressed as fractions of CO [24]. ^3^ The partition coefficients for diclazuril in different tissues were calculated based on the area method [25] through the previously reported diclazuril concentrations [1,19,21]. ^4^ This value is the sum of the hepatic artery plus portal vein flows. ^5^ The partition coefficients for diclazuril in the liver, lung, and rest compartment were optimized through the previously reported diclazuril concentrations [1,19,21]. ^6^ The plasma flow through the lung is equal to the cardiac output (Q_tot_); therefore, this value is equal to 1. ^7^ This value was calculated as 1 − (0.5712 + 0.2678 + 0.0064 + 0.0214 + 0.0071 + 0.0322 + 0.0161). ^8^ This value was calculated as 1 − (0.0764 + 0.2505 + 0.2012 + 0.2526).

**Table 4 animals-13-01512-t004:** The distribution information of those parameters subjected to Monte Carlo analysis.

Parameters	Unit	Average Value	SD	Min	Max	Source
CO ^1^	L/h/kg	9.88	2.07	7.81	11.95	[24]
BW ^2^	kg	1.5	0.15	1.35	1.65	[22]
Cl_he_	L/h/kg	0.00344	0.00002	0.00342	0.00346	Optimization
K_a_	1/h	0.1234	0.0007	0.1227	0.1241	Optimization
K_gut_	1/h	0.3838	0.0018	0.382	0.3856	Optimization
P_mu_ ^2^	unitless	0.1299	0.0129	0.117	0.1428	Area method
P_re_	unitless	1.2965	0.0064	1.2901	1.3029	Optimization
V_cmu_	%	57.12	14.73	42.39	71.85	[24]
V_csk_	%	13.38	2.82	10.56	16.2	[24]
V_cbl_	%	4.83	0.98	3.85	5.81	[24]
V_cfa_	%	13.4	2.01	11.39	15.41	[24]
Q_cmu_	%	7.64	1.14	6.5	8.78	[24]

^1^ It should be noted that the influential one was Q_clu_ (Appendix A), but because it had a fixed value (100% of cardiac output), its SD value was not available. So, we used the parameter of CO (cardiac output) here [24]. ^2^ Since we did not have access to actual SD values for the variables BW and P_mu_, we approximated them as 10% of their respective mean values.

**Table 5 animals-13-01512-t005:** Results of MAPE and the linear regression analysis between the predicted and observed diclazuril concentrations in plasma and tissues.

Reference	Tissues	Linear Regression Equation	The Determination Coefficient (R^2^)	MAPE (%)
[22]	Muscle	C_P_ = 0.8005C_O_ + 2.3109	0.9201	16.97
	Liver	C_P_ = 1.0471C_O_ − 6.0475	0.9839	8.19
[20]	Muscle	C_P_ = 0.9748C_O_ + 10.229	0.9801	8.03
	Liver	C_P_ = 0.9793C_O_ + 10.887	0.9968	2.94
	Plasma	C_P_ = 1.0283C_O_ + 33.409	0.9551	15.79
	Kidney	C_P_ = 1.119C_O_ − 36.708	0.9722	9.74
	Skin + fat	C_P_ = 0.8643C_O_ + 14.127	0.9816	14.00

C_O_ and C_P_ represent the observed and predicted diclazuril concentrations, respectively, in different tissues.

## Data Availability

The data that support the study findings are available upon request and after authorization by the authors.

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
