# Peer review of "Development and Application of a Physiologically Based Pharmacokinetic Model for Diclazuril in Broiler Chickens"

_animals, 2023, doi:10.3390/ani13091512_

Round 1

Reviewer 1 Report

This paper describes the development and validation of  PBPK model to assess the disposition of the anticoccidial drug, diclurazil in broilers and estimate a with residue withdrawal interval (WDI). The following are my concerns which should be addressed:

(1) The authors should be applauded for using another pharmacometric approach to estimate drug disposition for this drug which makes it unique. However, this should be stressed in the abstract and concluding statements.

(2) Please provide the label dose when you can. E.,g., Line 29 and line 59 need a label dose such mg/kg body weight and in the model structure section.

(3) Table 1 is  tad confusing; what is source of the data in the first line reporting a single oral dose of 1/ mg/kg BW by gavage? was it used in validation or optimization.

(4) It is a tad unfortunate that the modelers used acslxtreme software which is very much outdated and limited in flexibility but useful for this exercise. Authors would be advised to use R and R studio for future related work.

(5) Line 309: include "limited" after blood flow.

(6) Line 313, please provide a reference for area method.

(7) Happy that a sensitivity analysis was conducted, but would nice to mention explicitly that parameters assoLine 361ciated with the liver had no effect. Liver is often the target residue and the reader would be interested and as it had the highest tissue concentration.

(8) Please indicate the juisdication of the MRLs; with it is China, EU, Codex, etc. These values vary across juisdictions and would there influence the WDI computations.

Author Response

Dear reviewer,

We are truly grateful for your positive and constructive comments and suggestions on our manuscript entitled “Development and Application of a Physiologically Based Pharmacokinetic Model for Diclazuril in Broiler Chickens”. Based on these comments and suggestions, we have carefully modified the original manuscript. All changes made to the manuscript are marked up using the Track Changes function.

Our point-to-point responses to the comments are attached to this mail. We hope this revision meets Animals’ standards and receives some positive comments.

Yours sincerely,

Fan

Reviewer 2 Report

The manuscript presents a developed PBPK model which was applied for modelling of pharmacokinetics and diclazuril residues in broilers. The study was based on modelling of previously published data. The authors used contemporary software and the data were evaluated and whenever necessary, were adjusted with advanced software before their use in PBPK model.   The manuscript contains new data and its contribution to the field is significant. English language is at an acceptable level. The manuscript can be accepted for publication.

Introduction

The introduction is well written, without excessive information and the topic is precisely introduced.

Lines 55-57: The following sentence requires refinement: “The maximum residue limits (MRLs) of diclazuril in broiler products vary depending on the tissue type in China, which is 500, 1000, 2000, and 3000 μg/kg in muscle, skin + fat, kidney, and liver”. In this sentence, the place of “China” is not correct.

Materials and methods

The created PBPK model was clearly depicted in Figure 1.

Table 1 can be optimized. Look at line spacing.

Maybe it is good to add that the most important tissues which are investigated for residues were included in the model.

The model has a minor disadvantage because it was not taken into account that the transition time of the solution, from the crop to the intestines, if the crop is empty, is around 20 minutes (one sentence here, although a deep explanation was written in the discussion). This does not make the model less valuable, but an explanation why this lag time was not included in the model (Line 90). The amount of the drinking water/chicken is a bit high but an explanation was written.

Please, provide reference for the recommended Chinese method for calculation of withdrawal periods.

Results

The results are clearly presented and described. The supplementary file is helpful for evaluation of the suitability of the developed PBPK model for data simulation and prediction of the concentrations after multiple administration.

Table 5 should be moved after the paragraph with its first citation.

Discussion

The discussion shows deep knowledge of the authors on the published literature related to PBPK modelling in chickens. It is very well written and shows the advances of the model as well as its limitations. The discussion presents a very valuable comparison of the current model with the model developed for characterization of the pharmacokinetics of another antiprotozoal drug – monensin. The new developments with the current PBPK model were clearly pointed in comparative way with the existing information. Additionally, application of this model is well presented and illustrated.  

Author Response

(The authors gave the same response as above.)

Reviewer 3 Report

The physiologically based pharmacokinetic (PBPK) models are a reliable tool for forecasting veterinary drug residues in animal-derived foods. Previous PBPK models in chickens were mostly developed for antibacterials, and in this study, a PBPK model was first developed to aid in predicting diclazuril (an anticoccidial drug) concentrations in broilers and to further calculate the withdrawal periods of diclazuril. In general, this manuscript was designed correctly and is understandable. I recommend that it be considered for acceptance after some minor revisions listed below.

1. Abstract: In order not to mislead readers, the authors must also provide additional supplemental levels in both the water and feed. This review posits that the current 0-day withdrawal period may be influenced by the dosage of the supplement.

2. Lines 48 and 49: Besides China, have other countries approved the use of diclazuril in the full growth cycle of broilers?

3. Lines 57 and 58: It is important for the author to confirm whether the current zero-day withdrawal period is suitable for use in Europe, given the varying Maximum Residue Limit (MRL) values adopted by Europe. This will help ensure that the withdrawal period is appropriate for both China and Europe, and prevent any potential safety risks.

4. Lines 78-80: Chickens have a characteristic crop, which is a specialized digestive organ in birds that is used to store food before it is further processed in the stomach and intestines. But this anatomy was not included in the current model. Although incorporation of insignificant organs is a common practice in physiological pharmacokinetic modeling, authors are advised to describe it in brief terms.

5. Line 81: What are the characteristics of compounds that meet the blood flow rate-limiting law?

6. Lines 186-189: What software was used for the Monte Carlo analysis? Did the authors also use acslxtreme software?

7. Lines 247-248: Are there any reports available on the absolute bioavailability of diclazuril? Is there a difference in the results of this study? This model is both interesting and practical, and an accurate prediction of absolute bioavailability could facilitate further use of the PBPK model in veterinary drug development.

8. If there is no literature, what experiments can the authors conduct to establish the PBPK model to predict the residual amount of a drug in edible animal tissues.

Author Response

(The authors gave the same response as above.)
